

# CAMELS-CH: hydro-meteorological time series and landscape attributes for 331 catchments in hydrologic Switzerland

Marvin Höge[1], Martina Kauzlaric[2,13], Rosi Siber[1], Ursula Schönenberger[1], Pascal Horton[2,13], Jan Schwanbeck[2], Marius Günter Floriancic[3], Daniel Viviroli[4], Sibylle Wilhelm[2], Anna E. Sikorska-Senoner[5,6], Nans Addor[7,8], Manuela Brunner[3,9,10], Sandra Pool[11], Massimiliano Zappa[12], and Fabrizio Fenicia[1]

[1]Eawag, Dübendorf, Switzerland
[2]Geographisches Insitut, Universität Bern, Bern, Switzerland
[3]ETH Zürich, Zürich, Switzerland
[4]Universität Zürich, Zürich, Switzerland
[5]Federal Office of Meteorology and Climatology MeteoSwiss, Zurich-Airport, Switzerland
[6]Center for Climate Systems Modeling C2SM, ETH Zurich, Zurich, Switzerland
[7]Fathom, Bristol, UK
[8]University of Exeter, Exeter, UK
[9]WSL Institute for Snow and Avalanche Research SLF, Davos Dorf, Switzerland
[10]Climate Change, Extremes and Natural Hazards in Alpine Regions Research Center CERC, Davos Dorf, Switzerland
[11]University of Melbourne, Melbourne, Australia
[12]WSL, Birmensdorf, Switzerland
[13]Oeschger Centre for Climate Change Research, University of Bern, Bern, Switzerland

**Correspondence:** Marvin Höge (marvin.hoege@eawag.ch)

**Abstract.**

We present CAMELS-CH (Catchment Attributes and MEteorology for large-sample Studies - Switzerland), a large-sample hydro-meteorological data set for hydrological Switzerland in Central Europe. This domain covers 331 basins within Switzerland and neighbouring countries. About one third of the catchments are located in Austria, France, Germany and Italy. As an
Alpine country, Switzerland covers a vast diversity of landscapes, including mountainous environments, karstic regions, and several strongly cultivated regions, along with a wide range of hydrological regimes, i.e. catchments that are glacier-, snow-or rain-dominated. Similar to existing data sets, CAMELS-CH comprises dynamic hydro-meteorological variables and static catchment attributes.

CAMELS-CH (Höge et al., 2023, available at: https://doi.org/10.5281/zenodo.7957061) encompasses 40 years of data between 1st January 1981 and 31st December 2020, including daily time series of stream flow and water levels, and of meteorological data such as precipitation and air temperature. It also includes daily snow water equivalent data for each catchment starting from 2nd September 1998. Additionally, we provide annual time series of land cover change and glacier evolution per catchment. The static catchment attributes cover location and topography, climate, hydrology, soil, hydrogeology, geology, land use, human impact and glaciers. This Swiss data set complements comparable publicly accessible data sets, providing
data from the "water tower of Europe".



# 1 Introduction

Large sample hydro-meteorological data sets are essential to understand differences in catchment behavior and to improve hydrological predictions under changing conditions. An important objective of catchment hydrology is to learn about differences
between catchments to improve hydrological predictions in time, space and across scales (see also Andréassian et al., 2009). This pertains to, e.g., comparative hydrology (e.g. Falkenmark and Chapman, 1989), model regionalization (e.g. Parajka et al., 2005), catchment classification (e.g. Wagener et al., 2007), prediction in ungauged basins (e.g. Hrachowitz et al., 2013), and global trend analyses for hydro-meteorologic extremes and climate change impacts (Gudmundsson et al., 2019, 2021). The main types of required hydro-meteorological data include stream flow and its potential meteorological controls, such as precip-
itation, air temperature, potential evapotranspiration, and other auxiliary variables such as humidity, solar radiation and wind speed. Ideally, these data are complemented by maps or statistics of the catchment features that are believed to exert primary controls on the hydrological processes and the water cycle, such as topography, geology, soils, and land cover.

A seminal compilation of publicly available hydrologically-relevant data for a large sample of catchments is the MOPEX
data set that contains hydrologic and meteorological time series together with several static attributes for 438 catchments in the US (Duan et al., 2006; Schaake et al., 2006). This work paved the way for the data set created by Newman et al. (2015), who extracted atmospheric forcing from different meteorologic data products for 671 US catchments and combined this information with flow time series. Addor et al. (2017) complemented their data set with further static catchment attributes and created the first Catchment Attributes and MEteorology for Large-sample Studies (CAMELS) data set. It covers the continental USA and
features catchments with minimal to no human impact.

Recently, the availability of open hydro-meteorological data sets improved tremendously due to the publication of similar data sets. All of these cover a vast range of attributes for hundreds of catchments, typically within the national boundaries of respective countries or some in a transnational manner. Some introduced further attributes, e.g., on human impact per catch-
ment. The data set family was successively extended for: Chile (CAMELS-CL; Alvarez-Garreton et al., 2018), Great Britain (CAMELS-GB; Coxon et al., 2020), Brazil (CAMELS-BR and CABra; Chagas et al., 2020; Almagro et al., 2021, respectively), Australia (CAMELS-AUS; Fowler et al., 2021) and central Europe (mainly Austria) (LamaH-CE; Klingler et al., 2021). The latest development along these lines is Caravan (Kratzert et al., 2023), an initiative combining all existing CAMELS data sets as well as LamaH and HYSETS (Arsenault et al., 2020) with global forcing and attributes data, yielding a more spatially con-
sistent data set. Caravan also features a cloud-based platform enabling the fast extraction of forcing and attributes for any set of catchment polygons and the creation of Caravan extensions, e.g. for Denmark (Koch, 2022) and Israel (Efrat, 2023). Data sets for France (CAMELS-FR Andréassian et al., 2021) and Germany (CAMELS-DE Loritz et al., 2022) have been announced.

Large sample catchment data proved extremely valuable for facilitating research and collaboration. These data sets have fos-
tered many studies on large sample hydrology, focusing on model benchmarking, catchment classification, regionalization, and



prediction in ungauged basins (see Addor et al. (2020) for a review). Just to mention a few recent examples, they have been used to analyze and rank hydrologic signatures over hundreds of catchments (Addor et al., 2018), to benchmark machine learning models (Kratzert et al., 2019), to understand the mapping between catchment attributes and suitable model structures (Knoben et al., 2020), to investigate the inclusion of regional knowledge into hydrologic models (Gnann et al., 2021) or to regionalize

hydrologic model parameters for improved predictions in ungauged catchments (Pool et al., 2021). Further, CAMELS data were used to understand the correspondence between suitable model structures and observed hydro-meteorological signatures (David et al., 2022) and to demonstrate the performance of hybrid hydrologic models that combine physical principles with machine learning (Höge et al., 2022).

Open data sets have proven to be a remedy for several issues that hindered hydrological research in the past. Among others, these are problems related to economics and ownership (Gupta et al., 2014), to difficulties in the process of searching, accessing and using relevant data, which caused redundant efforts, and limited research advances (e.g. Viglione et al., 2010; Beniston et al., 2012), or to basic scientific principles like reproducibility. Large-sample hydrology and, in particular, global analyses were often hampered in the past and it took initiatives like the compilation of stream flow indices for several thousand basins

all over the globe (Do et al., 2018; Gudmundsson et al., 2018) or the first CAMELS data set for stream flow time series and additional data (Addor et al., 2017) to overcome such challenges and pave the way forward. At the scientific level, open research data initiatives are increasingly encouraged and previously unavailable data are becoming accessible through initiatives like this one. Like the other large-sample hydrology data sets before, CAMELS-CH follows the "FAIR Guiding Principles for scientific data management and stewardship" (Wilkinson et al., 2016).

## 2    Motivation

Within the European water cycle, Switzerland is sometimes referred to as the "water tower of Europe" (Viviroli and Weingartner, 2004; Viviroli et al., 2004), being located in the middle of the western continent and being home to the largest share of the entire Alpine glacier mass. Major European rivers such as the Rhine, Danube (Inn), Po (Ticino) and Rhone have their headwaters in Switzerland and flow to the neighbouring countries in the North (Germany), East (Austria), South (Italy) and West

(France), respectively. Hydrological Switzerland extends beyond the national boundaries and therefore comprises catchments both within political Switzerland and its vicinity, as shown in Figure 1 (a).

Yet, this integral part of the hydrologic system of Europe is subject to marked changes. Naming just a few, there is a significant change in the local flora as "Alps turn more green" (Rumpf et al., 2022), pronounced snow cover reduction with more than $8\%$ per decade on average since the early 1970s (Matiu et al., 2021), and a loss of glacier mass that amounts to nearly

$50\%$ since 1931 with nearly $30\%$ since 1980 alone (Mannerfelt et al., 2022). Stream flow patterns are changing, especially in mountainous basins, e.g., showing an increase in winter and spring discharge due to increasing temperatures (Birsan et al., 2005). Model-based hydrologic projections indicate a seasonal shift of the hydrologic cycle with decreasing summer runoff to increasing winter runoff for the coming decades (Köplin et al., 2012; Addor et al., 2014) with most drastic impacts in high-





elevation regions (Muelchi et al., 2021). Looking at hydrologic extremes, it was shown that parts of Switzerland experienced
a high frequency of flood events since about 1970 (Schmocker-Fackel and Naef, 2010). Projections indicate further changes
in flood seasonality and increasing flood magnitudes, yet mostly decreasing flood volumes for the future (Köplin et al., 2013;
Brunner et al., 2019a). Similarly, drought magnitudes are expected to change as a result of changes in their seasonality and
generation processes, such as increasingly frequent snowmelt deficits (Brunner et al., 2023).

All of these changes are direct effects of climate change that impact Switzerland and the water cycle of Europe in various
ways. In Switzerland, both natural and human-made systems are often still adapted to an overall abundance of water and to
seldom extremes throughout all seasons. Prospectively, they will have to adapt or be adjusted to the new conditions such as
more frequent floods or water shortages (see, e.g. Lanz, 2021). Climate change is likely to exert an heterogeneous impact on
the water cycle, given the diversity of the Swiss landscape, ranging from the karstic regions in the North-West to the Alps
in the South. Therefore, understanding the local impacts of climate change requires fine granularity studies that account for
local catchment properties. Catchment-specific hydro-meteorological, land use, snow and glacier data that cover decades are
an important prerequisite to understand and predict the evolution of these changes and are considered to be important assets
for climate impact assessments (e.g. Köplin et al., 2013).

With its 40 years of data, CAMELS-CH encompasses historic extreme events like the intense flooding in 2005 (Bezzola and
Hegg, 2007), the snow-melting-enforced flooding in 1999 (Forster and Hegg, 2000) or the major drought of 2018 (Brunner
et al., 2019b). Therefore, it represent an important resource to enable the investigation of hydrological regimes, potentially
under changing conditions, and for addressing other issues related to the water cycle and water resources management.

For comparability to and compatibility with similar former data sets, CAMELS-CH resembles the general structure of
already existing CAMELS data sets. We followed the structure of the British data set (cf. Coxon et al., 2020) to a large
extent because of similarities in the presence of, e.g., many populated basins, hydropower or reservoirs. In addition, we added
information on snow coverage, glaciation and geology to account for the specifics of an Alpine region.

## 3  Data sources and providers

CAMELS-CH is a compilation of data from various sources. First, data were collected from Swiss federal agencies, namely
the Federal Office for the Environment (FOEN, 2023), the Federal Office of Topography (swisstopo), the Swiss Federal Office
of Energy (SFOE), and the Swiss Federal Office of Meteorology and Climatology (MeteoSwiss, 2023). Second, further data
were obtained from Swiss research institutions and programs, i.e. the Swiss glacier inventory (GLAMOS, 2016) or the WSL
Institute for Snow and Avalanche Research (SLF, 2023) , with WSL being the referring to the Swiss Federal Institute for Forest,
Snow and Landscape Research (WSL, 2023). Third, these data were complemented by resources from the European Union
(EU) like different open-access data sets from Copernicus, the earth observation program of the EU, or by basin information
from local agencies of the EU neighbour countries around Switzerland (specified in Section A1.1). Respective hydrologic
data were collected and daily stream flow time series were aggregated from hourly data by and are available from Kauzlaric
et al. (2023). Fourth, we complemented observation-based data by well-established and validated simulation-based products,





i.e. hydro-meteorological time series simulated by the Precipitation-Runoff-Evapotranspiration Hydrotope (PREVAH) model (Viviroli et al., 2007), a distributed hydrological catchment modelling system that has been widely applied in Switzerland and other mountainous environments world-wide (e.g. Viviroli et al., 2009).

CAMELS-CH consists of catchment delineations, catchment time series with daily and annual resolution and static catchment attributes. These different data products are described in the following Sections 4, 5 and 6, respectively. Direct links to all data resources that are available online are provided in the Supplementary material section A1.

## 4  Catchments

CAMELS-CH comprises 298 river catchments and 33 lakes. While hydrological Switzerland covers many more surface water
bodies, the data set includes only those surface waters that are monitored by the FOEN (2023). That is, at this stage, it does not include any data from the Swiss Cantons, i.e. the member states of Switzerland. Out of the total 298 catchments, 195 are located within the national boundaries of Switzerland (and the small state Liechtenstein, which is also covered by the FOEN). Most of the 33 lakes are inland waters with only four extending to neighbouring countries (Lake Geneva, Lake Maggiore, Lake Lugano, Lake Constance). We decided to include lakes in addition to river catchments to allow for more comprehensive
analyses of the Swiss water cycle amid the high abundance of lakes in the country. With a few important exceptions, the lakes are regulated, i.e. their water level is managed (see Section A1.4 for further information). The other 103 catchments in the data set are located in the four neighbouring EU countries (see Figure 1 (a)): Austria (34 catchments), France (32), Germany (26) and Italy (11) and are not monitored by the FOEN. Out of the Austrian catchments, 15 coincide with those in LamaH-CE (see Section A1.1) which in turn includes parts of Switzerland and catchments of other countries along the Danube river system.

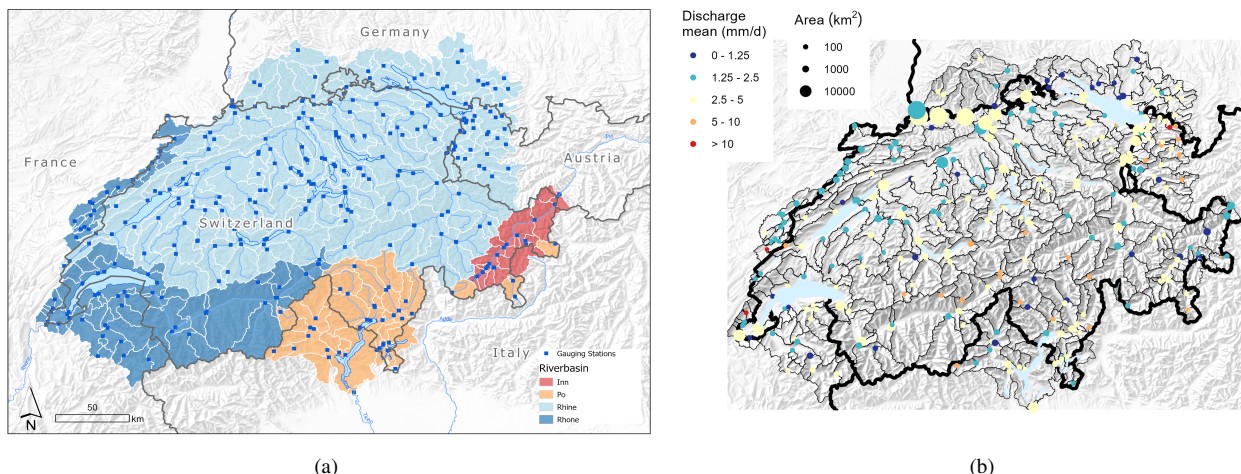

(a)                                              (b)

**Figure 1.** (a) Map of hydrologic Switzerland, covering political Switzerland and extending to neighbouring countires, with all gauging stations of CAMELS-CH and delineations of the four major river basins of Inn, Po, Rhine and Rhône; (b) Mean daily specific discharge and upstream catchment area at gauging stations (lake and river gauging stations with only water level recordings not shown.)



All catchment delineations within CAMELS-CH are provided in the Swiss coordinate system LV95 (sometimes also referred to as CH1903+). Those catchments within political Switzerland are based on the topographic delineation of catchments for Swiss water bodies by the FOEN (FOEN, 2020c, a). Note that, according to the FOEN, catchment outlet locations might not be fully identical to the gauging station positions. Yet, these are only minor discrepancies that do not restrict the usability of catchment-specific data series and the flow time series measured at the gauging stations. Catchment shapes outside of Switzerland are based on the EU-Hydro River Network database (EU-Hydro, 2020) and, when available, were complemented by geographic information from the same local agencies that provided discharge data (see Section 3).

Catchments sizes range over several orders of magnitude, from 0.6 km$^2$ for the Sperbelgraben brook (gauge_id: 2282) to about 36400 km$^2$ for the Rhine river (at gauge_id: 2613). Many catchments are nested, hence their discharge time series might show high correlation. In order to analyze their hierarchy, we provide two attributes (h1 and h2) that are described and demonstrated for the example of the Thur river (at gauge_id: 2044) in Appendix Section B.

All catchment delineations are provided as shape files in CAMELS-CH. Similarly to other hydro-meteorological data sets mentioned above, the catchment delineations were used to aggregate basin-specific meteorologic forcing data grids and spatially distributed attributes. Note that, despite thorough inspection and manual adjustments where needed, errors due to inaccurately defined catchment boundaries (in particular for the smallest catchments) cannot be fully excluded and might pertain to any analysis conducted using CAMELS-CH.

## 5 Time series

### 5.1 Daily time series

CAMELS-CH comprises an observation-based and a simulation-based set of hydro-meteorological time series as specified in Table 1 for a period from 1. January 1981 to 31. December 2020. Observed discharge and water level data were directly available from FOEN as time series recorded at the gauging stations. Yet, FOEN notes that data between 1. January 2019 and 31. December 2020 have not yet been validated at the time of their release in CAMELS-CH. Therefore, the observed data for these two years is provisional although regular quality assessments minimize potential shortcomings. Hydrologic data for catchments outside of political Switzerland were provided as hourly time series by Kauzlaric et al. (2023) and were averaged to daily resolution. Further research on hydrometry and uncertainty of observed discharge time series in Switzerland is discussed and referenced in the Appendix (Section 7.4).

Gridded observation-based data at daily resolution for precipitation, temperature and relative sunshine duration were obtained from MeteoSwiss. The spatial grid resolution is 2 km and each variable was aggregated using the catchment delineations introduced in Section 4, i.e. averaging precipitation, minimum, mean and maximum temperature and relative sunshine duration per catchment. The relative sunshine duration (rel_sun_dur) refers to the ratio between the effective sunshine duration (due to clouds or fog) and the maximum possible sunshine duration for a specific location on a certain day of the year. According to MeteoSwiss, sunshine is defined as solar irradiance $\geq 200$ W m$^{-2}$ and it is therefore used as a proxy for shortwave radiation





intensity.

Snow plays a crucial role in the hydrological cycle of Switzerland. Observation-based snow water equivalent (SWE) data were provided by the WSL Institute for Snow and Avalanche Research (SLF). For information about the snow cover model and data assimilation method, please refer to Magnusson et al. (2014). Basin-specific time series were spatially aggregated from a simulated snow data grid with a 1 km resolution. The simulations were based on 345 monitoring stations and the high-resolution daily precipitation product by MeteoSwiss (RhiresD version 2, see Section A1.2). Note that snow water equivalent

data are available only since 1. September 1998 and therefore do not cover the entire temporal extent of CAMELS-CH. Time series for basins smaller than 10 km$^2$ were removed per SLF recommendations due to uncertainties in the generated SWE estimates. Above 3000 m a.s.l., no monitoring stations are available and SWE data in high-elevation basins are subject to large uncertainties and have to be treated accordingly when used in analyses and modeling tasks.

The observed time series data above are complemented by simulation-based time series products to bridge the spatial dis-

crepancy between political and hydrological Switzerland, to fill temporal gaps in observation-based time series and to provide further meteorological variables that are not obtainable as observation-based products. Therefore, PREVAH model (Viviroli et al., 2009) simulations provide additional discharge time series and further simulated meteorologic variables are based on the PREVAH-internal WINMET tool that is used to pre-process and interpolate input variables for PREVAH. The observation-based meteorologic data by MeteoSwiss presented above served as input to WINMET, the simulated output was obtained on a

spatial grid with 500 m resolution. The final time series were aggregated per basin in the same way as the observed data from MeteoSwiss. PREVAH computations were conducted on the computational cluster ("Hyperion") of the WSL (2023), i.e. 56 computational nodes each with 20 cores at 2.2 (3.1) GHz (1120 nodes total) and 64 GB of RAM. The simulation-based time series were obtained similarly to the data set by Zappa and Brunner (2019) that was used and described extensively in Brunner et al. (2019b). Yet, while there 307 medium-sized catchments in Switzerland and the time between 1981 and 2018 were

considered, here we cover the 331 CAMELS-CH basins as defined in Section 4 between 1. January 1981 and 31. December 2020

All simulation-based variables are, of course, subject to assumptions and uncertainties introduced, e.g., by the chosen interpolation methods for meteorological variables (see Table 1 in  Viviroli et al., 2009), i.e. inverse distance weighting together with elevation-dependent regression. Yet, the simulation-based data are hydrologically validated (cf. presented in  Speich et al.,

2015) and can serve as a benchmark for other modeling studies. Besides simulated streamflow, precipitation and temperature that are also available as observation-based time series, we provide simulation-based estimates only for wind speed and relative humidity as well as potential and actual evapotranspiration (ET). Potential ET is based on the Penman-Monteith equation without interception correction (cf. Gurtz et al., 1999) and adjusted to actual (real) ET using local moisture, soil and vegetation conditions (for details see the referenced PREVAH model description in Section A1.3. Further, simulated radiation-related

variables (radiation_sim in Watts per square meter and sun_duration_sim in hours) complement the observation-based relative sunshine duration. Finally, two interception estimates are provided to describe the water balance at the soil surface underneath a plant canopy using so-called intercepted evapotranspiration (intercept_et_sim) and intercept storage (intercept_storage_sim)



that are further specified in Section A1.3). The data sources column in Table 1 shows the exact reference of each hydro-meteorologic variable and the ending "_sim" in the time series name indicates those data sets that are based on simulations.

**Table 1.** Catchment-specific hydro-meteorological variables available as daily time series in CAMELS-CH

| Time series class | Time series names | Description | Unit | Data Source |
|---|---|---|---|---|
| Hydrological time series (1. Jan. 1981 - 31. Dec. 2020) | discharge_vol | observed catchment discharge | $m^3 \ s^{-1}$ | Swiss Federal Office for the Environment (FOEN, 2023) and neighbouring countries resources as specified in Section A1.1 |
| | discharge_spec | observed catchment-specific discharge (converted to millimetres per day using catchment areas described in Sect. 6.1) | $mm \ d^{-1}$ | |
| | waterlevel | observed daily water head above sea level | m | |
| | discharge_vol_sim | simulated daily averaged absolute discharge | $m^3 \ s^{-1}$ | Precipitation-Runoff-Evapotranspiration Hydrotope (PREVAH) model (Viviroli et al., 2007) |
| | discharge_spec_sim | simulated daily averaged discharge normalized by catchment area | $mm \ d^{-1}$ | |
| Meteorological time series (1. Jan. 1981 - 31. Dec. 2020) | precipitation | observed daily summed precipitation | $mm \ d^{-1}$ | Swiss Federal Office of Meteorology and Climatology (MeteoSwiss, 2023) |
| | temperature_min | observed daily minimum temperature | °C | |
| | temperature_mean | observed daily averaged temperature | °C | |





| Time series class | Time series names | Description | Unit | Data Source |
|---|---|---|---|---|
| | temperature_max | observed daily maximum temperature | °C | |
| | rel_sun_dur | observed daily averaged relative sunshine (solar irradiance $\geq 200$ W m$^{-2}$) duration | % | |
| | swe | observed daily averaged snow water equivalent | mm | Swiss WSL Institute for Snow and Avalanche Research (SLF, 2023) |
| | precipitation_sim | simulated daily averaged precipitation | mm d$^{-1}$ | Precipitation-Runoff-Evapotranspiration Hydrotope (PREVAH) model (WINMET tool; Viviroli et al., 2007) |
| | temperature_sim | simulated daily averaged temperature | °C | |
| | radiation_sim | simulated daily averaged global radiation | W m$^{-2}$ | |
| | sun_duration_sim | simulated daily averaged sunshine duration | h | |
| | wind_sim | simulated daily averaged wind speed | m s$^{-1}$ | |
| | rel_humidity_sim | simulated daily averaged relative humidity | % | |
| | pet_sim | simulated daily averaged potential evapotranspiration (Penman-Monteith equation without interception correction) | mm d$^{-1}$ | |
| | et_sim | simulated daily averaged actual evapotranspiration | mm d$^{-1}$ | |



| Time series class | Time series names | Description | Unit | Data Source |
|---|---|---|---|---|
| | intercept_et_sim | simulated daily averaged intercepted evapotranspiration | mm d$^{-1}$ | |
| | intercept_storage_sim | simulated daily averaged intercept storage | mm | |

## 5.2 Annual time series

Glaciation and land cover have been undergoing considerable changes in certain basins over the last decades. Therefore, CAMELS-CH includes annual time series of both glacier and land cover attributes as specified in Table 2 for each catchment.

### 5.2.1 Glaciers

Glacier data within Switzerland are based on the Swiss Glacier inventory GLAMOS (Linsbauer et al., 2021) that is updated every few years. The attributes provided are interpolated linearly between the GLAMOS versions of 1973 and 2016. GLAMOS comprises data on glacier volumes and areas as well as shapefiles for each individual glacier. These glacier outlines were intersected with the catchment delineations (see Section 4) in order to aggregate the glacier volume and to obtain the glaciated area of the catchment.

For catchments in neighbouring countries only the temporal development of glacier area is available based on the pan-European glacier inventories of the years 2003 and 2015 by Paul et al. (2011) and Paul et al. (2019); Paul et al. (2020) that cover the entire Alpine region. Note several national glacier inventories were established as for France in 2006-2009 (Gardent et al., 2014) and Austria in 2012 (Fischer et al., 2015). However, to provide the best possible temporal and methodological consistency, pan-European glacier inventories were used. The glacier inventory from 2003 is based on Landsat (Thematic Mapper, 30m resolution). Sentinel-2 (S2, 10m resolution) satellite data and former national glacier inventories as guidance to map the glacier outlines for the pan-European inventory of 2015 (see Paul et al., 2011, 2020). Note that there is also the Randolph glacier inventory (Pfeffer et al., 2014) available that is based on Landsat satellite data from 2003 and was used for CAMELS-CL (Alvarez-Garreton et al., 2018). Yet, while this might be a valuable resource for global analyses and certain regions, it is not the most accurate resource for the European Alps and it was therefore not used here.

The more recent glacier inventory of 2015 by Paul et al. (2020) is stated to be more accurate than the older inventory of 2003 due to the more precise data and more experience in mapping glacier outlines. This means, however, that certain glaciers present a slightly larger area in 2015 than in 2003 and appear to have grown over the years - opposite to the overall confirmed trend of glacier decline (Mannerfelt et al., 2022). This artifact is countered by transferring the area loss rates of Swiss glaciers of similar extent in GLAMOS to the affected glaciers. Starting from the year 2015, this rate was then used to estimate the areal development backward to 1980 and forward to 2021 for glaciers that apparently but unrealistically have grown. We acknowledge that this methodology might introduce further uncertainty because it is a strong simplification as it does not



take into account glacier exposition, basin morphology, regional climate, etc. Yet, we consider GLAMOS data to be a highly accurate reference and to provide representative and, therefore, transferable trends for the evolution of glaciers in the European Alps.

CAMELS-CH provides annual time series for glacier area (glac_area), glacier mass (glac_mass) and glacier volume (glac_vol)
for all glaciers within political Switzerland (see Table 2). In order to allow for relating Swiss glacier area to volume and mass during investigations, we decided not to aggregate the glacier areas from GLAMOS and from the pan-European glacier inventories for catchments that cross the borders of Switzerland. For these catchments and those that lie entirely outside of political Switzerland, we provide glacier areas as a separate variable glac_area_neighbours that can manually be aggregated with the GLAMOS-based glacier area.

### 240  5.2.2  Land cover

Land use changes were estimated based on the CORINE Land Cover (CLC) data sets (Büttner et al., 2004) from the European Copernicus program, with CORINE abbreviating Coordination of Information on the Environment. This remote sensing program monitors - among many other things - the land cover distribution (e.g., urban, forest types, agriculture types) since 1985 over the larger European area. From this geo-referenced data, we aggregated areal percentages of each land cover type for all
catchments. The accuracy of the CLC files is 100 m or better, with improving accuracy over the decades since the first CLC release. CLC data sets exist for the years 1990, 2000, 2006, 2012 and 2018. We applied linear interpolation between these time points to fill the years in between and repeated the value 2018 to fill in the last years. While the first CLC data set was released in 1990, data for Switzerland is only available since 2000 when the country joined the Copernicus program. Hence, for Swiss basins, annual data are available between 2000 and 2021 and for the neighbouring countries' basins, the annual time series
start from 1990.

CAMELS-CH covers land cover percentages on the following 12 categories: agriculture, forest (coniferous, deciduous and mixed), grass and herb vegetation, scrub vegetation, wetlands, ice and perpetual snow, inland water surface, rock (loose and solid), settlements/urban and unknown/blank. Note that there might be slight deviations between the values derived from the glacier data sources (see Section 5.2.1) with respect to the percentage of catchment area covered by ice and perpetual snow
(ice_perc). This is due to deviating assessment techniques - CLC is only based on satellite data that might not distinguish as clearly between glacier and snow surfaces as it was done for the glacier inventories. Therefore, we recommend using the glacier attributes for respective applications

## 6  Catchment attributes

### 6.1  Location and topography

Location attributes were derived from data provided by the FOEN and comprise basic information about the country each gauging station is located in, its name and coordinates as well as the corresponding water body name and type (river or





lake). As identifiers, we provide the unique four-digit gauge_id number in accordance with the FOEN notation and a six-letter identifier (id6) that is derived from water body name and gauge site name. The unique gauge_id numbers coincide with the numbers used by the FOEN for basins within political Switzerland starting with "2". Correspondingly, we enumerated

gauge_id numbers for Austria, France, Germany and Italy starting with "3", "4", "5" and "6", respectively. This enumeration is CAMELS-CH specific, but all catchments can be uniquely identified in the data source by Kauzlaric et al. (2023) using the id6 identifier.

Topographic attributes include gauging station elevation elevation, catchment area and statistical means of elevation and slope distributions of each basin. They were partially also provided by the FOEN and swisstopo or derived from the EU

digital elevation model (25 m resolution; EU-Hydro, 2020) to have a unique source of information for Switzerland and the surrounding countries. For the catchments within political Switzerland, the obtained attributes were compared to swisstopo data, which revealed nearly identical estimates for the catchment-aggregated attributes provided in CAMELS-CH.

Further attributes on location and topography, e.g., catchment hierarchy or further specifics of hydrological Switzerland covering several countries, are provided in Appendix Section B.

**Table 2.** Catchment-specific static attributes available in CAMELS-CH ($^*$Soil attributes: each soil type/property is accompanied by percentiles (5, 25, 50, 75, 90), distribution skewness and missing percentage across the catchment) with additional annual time series available for glacier and land cover attributes (see Section 5.2)

| Attribute class | Attribute name | Description | Unit | Data Source |
|---|---|---|---|---|
| Location and topography | gauge_id | catchment identifier according to FOEN notation, adjusted to neighbouring countries | – | Swiss Federal Office for the Environment (FOEN, 2023) and neighbouring countries resources as specified in Section A1.1 |
| | country | country of gauging station | – | |
| | gauge_name | gauging station name | – | |
| | water_body_name | water body name | – | |
| | id6 | identifier based on gauging station and water body names | – | |
| | water_body_type | water body type (stream or lake) | – | |
| | gauge_lon | gauging station longitude | ° | |





| Attribute class | Attribute name | Description | Unit | Data Source |
|---|---|---|---|---|
| | gauge_lat | gauging station latitude | ° | |
| | gauge_easting | gauging station easting | m | |
| | gauge_northing | gauging station northing | m | |
| | gauge_elevation | gauging station elevation | m.a.s.l. | |
| | area | catchment area | km$^2$ | |
| | elev_mean | mean elevation within catchment | m.a.s.l. | Swiss Federal Office of Topography (swisstopo, 2015) / EU digital elevation model (v1.1; EU-DEM, 2016) |
| | elev_min | minimum elevation within catchment | m.a.s.l. | |
| | elev_percentiles | elevation percentiles (10, 25, 50, 75 and 90 %) | m.a.s.l. | |
| | elev_max | maximum elevation within catchment | m.a.s.l. | |
| | slope_mean | catchment mean slope over all grid cells | ° | |
| | flat_area_perc | percentage of catchment area with slope smaller than 3° | % | |
| | steep_area_perc | percentage of catchment area with slope greater than 15° | % | |
| Climate | ind_start_date | start date for indices calculation | – | MeteoSwiss (2023, observation-based attributes) / PREVAH (WINMET tool; Viviroli et al., 2007, simulation-based attributes) |
| | ind_end_date | end date for indices evaluation | – | |





| Attribute class | Attribute name | Description | Unit | Data Source |
|---|---|---|---|---|
| | ind_number_of_years | number of years for indices evaluation | – | |
| | p_mean | mean daily precipitation | mm d$^{-1}$ | |
| | pet_mean | mean daily potential evapo-transpiration (PET; Penman-Monteith equation without interception correction) | mm d$^{-1}$ | |
| | aridity | aridity (ratio of mean daily PET to mean daily precipitation) | – | |
| | p_seasonality | seasonality and timing of precipitation (estimated using sine curves to represent the annual temperature and precipitation cycles, positive (negative) values indicate that precipitation peaks in summer (winter), and values close to zero indicate uniform precipitation throughout the year). See Eq. (14) in Woods (2009)) | – | |
| | frac_snow | fraction of precipitation falling as snow, i.e. while temperature is $< 0°C$ | – | |
| | high_prec_freq | frequency of high-precipitation days ($\geq 5$ times mean daily precipitation) | d yr$^{-1}$ | |





| Attribute class | Attribute name | Description | Unit | Data Source |
|---|---|---|---|---|
| | high_prec_dur | average duration of high-precipitation events (number of consecutive days $\geq 5$ times mean daily precipitation) | d | |
| | high_prec_timing | season during which most high-precipitation days occur, e.g. 'jja' for summer. If two seasons register the same number of events a value of NA is given. | season | |
| | low_prec_freq | frequency of dry days ($< 1$ mm $d^{-1}$) | d yr$^{-1}$ | |
| | low_prec_dur | average duration of dry periods (number of consecutive days $< 1$ mm $d^{-1}$ mean daily precipitation) | d | |
| | low_prec_timing | season during which most dry days occur, e.g. 'son' for autumn. If two seasons register the same number of events a value of NA is given. | season | |
| Hydrology | sign_start_date | start date for signature evaluation | – | FOEN (2023, observation-based attributes) / PREVAH model (Viviroli et al., 2007, simulation-based attributes) |
| | sign_end_date | end date for signature evaluation | – | |
| | sign_number_of_years | number of years for signature evaluation | – | |
| | q_mean | mean daily specific discharge | mm $d^{-1}$ | |



| Attribute class | Attribute name | Description | Unit | Data Source |
|---|---|---|---|---|
| | runoff_ratio | runoff ratio (ratio of mean daily discharge to mean daily precipitation) | – | |
| | stream_elas | stream flow precipitation elasticity (sensitivity of stream flow to changes in precipitation at the annual timescale, using mean daily discharge as reference, see Coxon et al. (2020) and reference therein) | – | |
| | slope_fdc | slope of the flow duration curve (between the log-transformed 33rd and 66th stream flow percentiles, see Coxon et al. (2020) and reference therein) | – | |
| | baseflow_index_landson | base flow index (see Coxon et al. (2020) and reference therein) | – | |
| | hfd_mean | mean half-flow date (number of days since 1. Oct at which the cumulative discharge reaches half of the annual discharge) | d | |
| | Q5 | 5 % flow quantile (low flow) | mm d$^{-1}$ | |
| | Q95 | 95 % flow quantile (high flow) | mm d$^{-1}$ | |



| Attribute class | Attribute name | Description | Unit | Data Source |
|---|---|---|---|---|
| | high_q_freq | frequency of high-flow days (> 9 times the median daily flow) | d yr$^{-1}$ | |
| | high_q_dur | average duration of high-flow events (number of consecutive days > 9 times the median daily flow) | d | |
| | low_q_freq | frequency of low-flow days (< 0.2 times the mean daily flow) | d yr$^{-1}$ | |
| | low_q_dur | average duration of low-flow events (number of consecutive days < 0.2 times the mean daily flow) | d | |
| | zero_q_freq | fraction of days with zero stream flow | – | |
| Soil* | sand_perc | percentage sand | % | EU-SoilHydroGrids (2017) / European Soil Database Derived (ESDD) data (ESDD, 2013) |
| | silt_perc | percentage silt | % | |
| | clay_perc | percentage clay | % | |
| | organic_perc | percentage organic material | % | |
| | coarse_fragm_perc | percentage coarse fragments | % | |
| | bulk_dens | bulk density | g cm$^{-3}$ | |
| | tot_avail_water | total available water content | mm | |
| | porosity | volumetric porosity | – | |
| | conductivity | saturated hydraulic conductivity | cm h$^{-1}$ | |





| Attribute class | Attribute name | Description | Unit | Data Source |
|---|---|---|---|---|
| | root_depth | depth available for roots | m | |
| Hydrogeology | unconsol_coarse_perc | well permeable gravel in valley bottoms | % | Hydrogeologic Maps of Switzerland (geo.admin.ch, 2016) and Germany (GDK, 2019) |
| | unconsol_medium_perc | permeable gravel outside of valley bottoms, sandy gravel, medium- to coarse-grained gravel | % | |
| | unconsol_fine_perc | loamy gravel, fine- to medium-grained debris, moraines | % | |
| | unconsol_imperm_perc | clay, silt, fine sands and loamy moraines | % | |
| | hardrock_perc | fissured and porous, non-karstic hard rock: conglomerates, sandstone, limestone with marl layers; crystalline rock: granite, granodiorites, tonalite | % | |
| | hardrock_imperm_perc | marl, shale, gneiss and cemented sandstone | % | |
| | karst_perc | carbonate rock: limestone, dolomite, rauhwacke; sulphate-containing rock: gypsum, anhydrite | % | |
| | water_perc | glaciers, firn, surface waters | % | |
| | null_perc | without defined hydrogeology | % | |





| Attribute class | Attribute name | Description | Unit | Data Source |
|---|---|---|---|---|
| | ext_area_perc | catchment areal percentage not covered by the data source | % | |
| Geology | geo_porosity | average catchment geologic porosity | – | GLHYMPS 2.0 (Huscroft et al., 2018) |
| | geo_log10_permeability | average logarithmic catchment geologic permeability | $\log_{10}(m^2)$ | |
| | unconsol_sediments | percentage of unconsolidated sediments | % | GLiM (Hartmann and Moosdorf, 2012) |
| | siliciclastic_sedimentary | percentage of siliciclastic sedimentary rocks | % | |
| | mixed_sedimentary | percentage of mixed sedimentary rocks | % | |
| | carbonate_sedimentary | percentage of carbonate sedimentary rocks | % | |
| | pyroclastic | percentage of pyroclastics | % | |
| | acid_volcanic | percentage of acid volcanic rocks | % | |
| | basic_volcanic | percentage of basic volcanic rocks | % | |
| | acid_plutonic geo_pa | percentage of acid plutonic rocks | % | |
| | intermediate_plutonic | percentage of intermediate plutonic rocks | % | |
| | basic_plutonic | percentage of basic plutonic rocks | % | |
| | metamorphics | percentage of metamorphics | % | |
| | water_geo | percentage of water bodies | % | |





| Attribute class | Attribute name | Description | Unit | Data Source |
|---|---|---|---|---|
| | ice_geo | percentage of ice and glaciers | % | |
| Glacier | glac_area_ch | glacier area of Swiss glaciers per catchment | km$^2$ | Swiss Glacier Inventory (GLAMOS, 2016) |
| | glac_vol_ch | glacier volume of Swiss glaciers per catchment | km$^3$ | |
| | glac_mass_ch | glacier mass of Swiss glaciers per catchment | MT ($10^6$ metric tons) | |
| | glac_area_neighbours | glacier area of glaciers attributed to neighbouring countries (Austria, France, Italy) | km$^2$ | Paul et al. (2019) |
| Land cover | crop_perc | percentage of agriculture | % | Copernicus: Corine Land Cover (CLC, 2000) |
| | grass_perc | percentage of grass and herb vegetation | % | |
| | shrub_perc | percentage of medium-scale vegetation | % | |
| | dwood_perc | percentage of deciduous forest | % | |
| | mix_wood_perc | percentage of mixed forest | % | |
| | ewood_perc | percentage of coniferous forest (evergreen) | % | |
| | wetlands_perc | percentage of wetlands | % | |
| | inwater_perc | percentage of inland water | % | |
| | ice_perc | percentage of glaciers and perpetual snow | % | |
| | loose_rock_perc | percentage of loose rocks and bare soils | % | |



| Attribute class | Attribute name | Description | Unit | Data Source |
|---|---|---|---|---|
| | rock_perc | percentage of hard rocks and bare soils | % | |
| | urban_perc | percentage of urban and settlements | % | |
| | dom_land_cover | dominant land cover type | – | |
| Human impact | n_inhabitants | population in catchment area | – | Geostat population grid (eurostat, 2018) |
| | dens_inhabitants | population density in catchment area | $km^{-2}$ | |
| | num_reservoir | number of reservoirs | – | Swiss Federal Office of Energy (SFOE, 2020) |
| | reservoir_cap | total storage capacity of reservoirs in megalitres | ML | |
| | reservoir_he | percentage of total reservoir storage used for hydroelectricity | % | |
| | reservoir_fs | percentage of total reservoir storage used for flood storage | % | |
| | reservoir_irr | percentage of total reservoir storage used for irrigation | % | |
| | reservoir_nousedata | percentage of total reservoir storage where no use data were available | % | |
| | reservoir_year_first | year the first reservoir was built | % | |
| | reservoir_year_last | year the last reservoir was built | % | |





| Attribute class | Attribute name | Description | Unit | Data Source |
|---|---|---|---|---|
| | hp_count | number of hydropower plants in the catchment with at least 300 kW installed capacity | – | Swiss Federal Office of Energy (SFOE, 2022) |
| | hp_qturb | sum of discharge capacity | m$^3$ s$^{-1}$ | |
| | hp_inst_turb | installed capacity | MW | |
| | hp_max_power | maximal bottleneck capacity | MW | |

## 6.2 Climate and hydrology

Climatic indices and hydrologic signatures were evaluated based on both observation- and simulation-based time series (see Table 1): For the climatic indices, precipitation and temperature from both MeteoSwiss (obs) and WINMET (sim) were used. Yet, potential evapotranspiration estimates are only simulation-based (pet_sim). For the hydrologic signatures, observation-based precipitation data from MeteoSwiss were used together with discharge data by the FOEN. The simulation-based equivalents were based on PREVAH and WINMET data for both variables. Note that observation-based hydrologic signatures are not available for the lakes because only water level is recorded at these gauging stations. Using PREVAH is was possible to generate discharge at these lake stations and hence hydrologic signatures are available for these simulated data.

For calculating both attribute categories, only complete observed hydrological years (1. October to 30. September of the following year) were used and up to 5% of missing values were tolerated per hydrological year. Note that observed discharge is the most limiting variable. Hence, if discharge was only available for three hydrologic years for a certain basin to allow for the calculation of hydrologic signatures, the climatic indices were evaluated only for the same three years for the sake of consistency. Simulated time series of all variables are complete for all catchments. Hence, simulation-based attributes can serve as substitution for missing observation-based ones. A comparison between attributes derived from both sources is conducted in Section 7.2.

In Table 2, we provide the same climatic and hydrologic attributes and the corresponding descriptions as in CAMELS-GB with the difference that we cover the period 1981–2020 and additionally report the start date, end date and number of years used for the calculations. For further information on specific attributes, see Coxon et al. (2020). Figure 1 (b) shows the mean observed discharge at each gauging station normalized by the respective catchment area with the dot size indicating the magnitude of catchment area. It can clearly be seen that at the boundaries of hydrologic Switzerland where the major rivers flow to neighbouring countries, the catchment areas are largest, resembling the encompassing catchments indicated in Figure 1 (a). For example, in the North, the gauging stations at the Rhine river cover about two thirds of the entire hydrologic Switzerland.





### 6.3 Soil attributes

The soil attributes listed in Table 2 were calculated using two data sources covering the entire spatial extent of hydrological Switzerland: The data set EU-SoilHydroGrids (3D soil hydraulic database of Europe) at 250m spatial resolution (Tóth et al.,

2017) and the European Soil Database Derived (ESDD) data at 1km spatial resolution (Hiederer, 2013a, b). Further, we provide soil attributes based on the global SoilGrids data set at 250m resolution (Hengl et al., 2017) in Appendix Section B. These include soil_depth as an additional variable and second estimates for several variables in Table 2. Yet, the variables coarse_fragm_perc, porosity and conductivity are only available from the more comprehensive European data sets which is why we selected these as primary data sources.

### 305 6.4 Hydrogeology

The hydrogeologic attributes as listed and described in Table 2 were primarily extracted from the hydrogeologic map of Switzerland (geo.admin.ch, 2016). This map covers the largest part of hydrological Switzerland with the exception of some parts in the North. Hence, a hydrogeologic map of Germany (GDK, 2019) was used in addition to the Swiss data, which extends the hydrogeologic coverage and significantly improves the coverage of the spatial extent of karstic regions in the West

and North. The remaining percentage per catchment not covered by any of the data sources is reported as 'external'. Further information is available in the Appendix (Section A1.5) and Section A2 offers a direct correspondence between the variables listed here and those in CAMELS-GB (Coxon et al., 2020).

### 6.5 Geology

The Swiss landscape is dominated by the different Alpine geologies that were derived from different data sources: Attributes

from GLiM (Global lithological map Hartmann and Moosdorf, 2012) indicated by lowercase abbreviations (geo_su, geo_ss,...) provide a general geologic categorization of rock type fractions (resolution at $0.5°$, ca. 1:3,750,000) and are typically sufficient for many hydrological applications. From the GLHYMPS 2.0 data set, (GLobal HYdrogeology MaPS 2.0; Huscroft et al., 2018) logarithmic permeability and porosity estimates are provided (average polygon size 100 km$^2$). Both GLiM and GLHYMPS-based variables were also used in the original CAMELS-US data set (Addor et al., 2017) which facilitates inter-

comparisons between data sets. Yet, note that with GLHYMPS version 2.0, a global surface mean logarithmic permeability is estimated to be $-12.7 \pm 1.7 \mathrm{m}^2$ which is about one order of magnitude higher than the one estimated in version 1.0 (Gleeson et al., 2014). This has to be considered when comparing the geologic variables from CAMELS-CH with other CAMELS data sets.

For hydro(geo)logic purposes, hydraulic conductivity $K$ (in m/s) rather than log10_permeability $k$ (in $\log_{10}(\mathrm{m}^2)$) might be

of interest. Using approximations of the density of water (1000 kg/m$^3$), gravitation (10 m/s$^2$) and the viscosity of water ($10^{-3}$ kg (m s)$^{-1}$), hydraulic conductivity can be estimated via: $K = 10^7 \cdot 10^k$ (see Gleeson et al., 2011).

Further attributes that allow for more comprehensive investigations regarding the geology of Switzerland and are based on Swiss data sources are provided in Appendix B.





## 6.6 Glaciers and land cover

In addition to respective annual time series in Sections 5.2.1 and 5.2.2, attributes for both glaciers and land cover are available as static attributes referring to the year 2000. If values from another year shall be used as static attributes, please refer to the annual time series per basin. Details about the glacier and land cover attributes and their data sources are given in Section 5.2.

## 6.7 Human Impact

Human impact attributes comprise the estimated population per catchment as well as data on reservoirs and hydropower.
In order to have a unique data source for all catchments, population was derived from the EU Geostat data set (1km$^2$ grid resolution; eurostat, 2018) that also covers Switzerland. Reservoir (SFOE, 2020) and hydropower (SFOE, 2022) data were obtained from the Swiss Federal Office of Energy (SFOE) and only cover catchments within political Switzerland. Swiss reservoirs are mainly used for flood control, hydro electricity and water supply. The hydropower plants supervised by SFOE have an installed capacity of at least 300 kW, smaller once are not included here.

## 7 Data analysis and Discussion


### 7.1 Annual time series: Glacier areas

Within the family of CAMELS datasets, it is a novelty of CAMELS-CH to provide annual time series data for glaciers and land cover. Therefore, we demonstrate exemplarily which further types of analysis are thereby enabled: Using the annual time series, it is possible to allocate glacier decline catchment-specifically between 1980 and 2020 in the Alps of hydrological
Switzerland. Figure 2 (a) shows the areal extend of glacier per (sub-)catchments in 1980 and (b) clearly depicts the linear trend of glacier area loss over the four decades with 100% referring to the glacier extend in 1980 as reference level. All glaciers show considerable decline. Yet, while some glaciers in basins disappeared nearly or entirely, others have lost less.



**Figure 2.** Glacier area evolution between 1980 and 2020 with (a) Glacier area per (sub-)catchment in 1980; (b) relative individual declines and area-weighted mean (turquoise) over 40 years for all (sub-)catchments with reference year 1980; (c) and (d) relative glacier area losses per catchment in 2000 and 2020, respectively, since 1980.

Figure 2 (c) and (d) show the relative glacier loss in the years 2000 and 2020, respectively, and indicate that basins in the eastern Swiss Alps suffer from greater decline. It can be seen, that those basins where glaciers have nearly or fully disappeared are located at the margins of the Alps which are generally smaller glaciers. The largest glaciers, located at high elevations in the western Alps, show losses up to 20% in 2000 and up to 40% in 2020. Yet, their masses have declined even more strongly with their areas being typically restricted by mountain ridges. Therefore, they loose height and shrink in volume and mass more than in area.

### 7.2 Hydrologic attributes from observed vs. simulated data

The observed mean specific discharge together with the area upstream to the respective gauging stations are shown in Figure 1 (b). The largest major catchment of the Rhine river that covers large parts of hydrologic Switzerland is represented by



the gauging stations in the North. The other three major rivers that flow to surrounding parts of Europe (Rhone, Po and Inn) yield large dots at the respective gauging stations in the other three cardinal directions. Within Switzerland, the gauging stations typically cover smaller catchments. Most gauging stations depict a mean specific discharge in the medium category 360 $2.5 - 5\mathrm{mm\,d^{-1}}$, with no general spatial pattern being visible for all categories.

Figure 3 depicts mean half-flow date generated from both observation-based (a) and simulation-based data (b). It nicely shows how in high-altitude, snow- (and in some cases glacier-) dominated catchments, it takes many more days since the beginning of the hydrologic year (always 1. October) until the cumulative discharge reaches half of the entire annual discharge, compared to low altitude basins that are more rain-dominated. The overall resemblance between the hfd_mean generated from 365 observations and simulations is very high. Therefore, in such a case simulation-based attributes are suitable to complement observation-based attributes that might have gaps in certain catchments.

For observation-based and simulation-based runoff_ratio in Figure 3 (c) and (d), respectively, this is not as clearly visible because the simulation-based catchment indices show overall higher values, with the highest values concentrated in the mountains. Runoff ratios larger than one typically indicate, e.g. external inputs of water like extracted groundwater, water release for 370 hydropower production, melting of snow or glaciers that are all not explicitly metered or publicly available. In mountainous regions, the estimation of the water balance is often challenging because water from mountains provide large (temporarily stored) fractions of precipitation to the total discharge of catchments in their vicinity (Viviroli and Weingartner, 2004). Further, precipitation data can cause runoff ratios larger than one by mainly two reasons: gauge undercatch (e.g., Kochendorfer et al., 2017) and, in particular for high-elevation catchments, a poor representation of the precipitation gauge network (e.g., Viviroli 375 et al., 2011). A general estimation of accuracy and uncertainty in the precipitation data by MeteoSwiss can be found in the respectively referenced product sheet in Section A1.2.

Aridity values in 3 (e, f) are between 0 and 1 and indicate that catchments in hydrologic Switzerland are energy-limited (aridity < 1) rather than water-limited (aridity > 1) which is expected. Simulation-based aridity in Figure (f) shows a more even distribution that the observation-based attribute in (e) with less values towards the extremes 0 and 1. For example, they 380 do not reflect the lowest aridity values in high altitudes. Therefore, whenever users of CAMELS-CH intend to complement observation-based with simulation-based attributes (hydrologic and likewise climatic), we recommend a thorough attribute-specific comparison and to acknowledge potential discrepancies.

**Figure 3.** Comparison of observation-based (left) and simulation-based (right) hydrologic signatures: mean half-flow date (a,b) and runoff ratio (c,d), and climate index: aridity (e,f)

## 7.3 Bio-geographic regions

Regions in Switzerland can be distinguished according to their climate, flora and fauna, topography and/or geology. Political
Switzerland comprises six of these so-called bio-geographic regions (see Section A1.6) and CAMELS-CH extends these to the
neighbouring countries, adding also the southern Black Forest as seventh region as shown in Figure 4 (a).

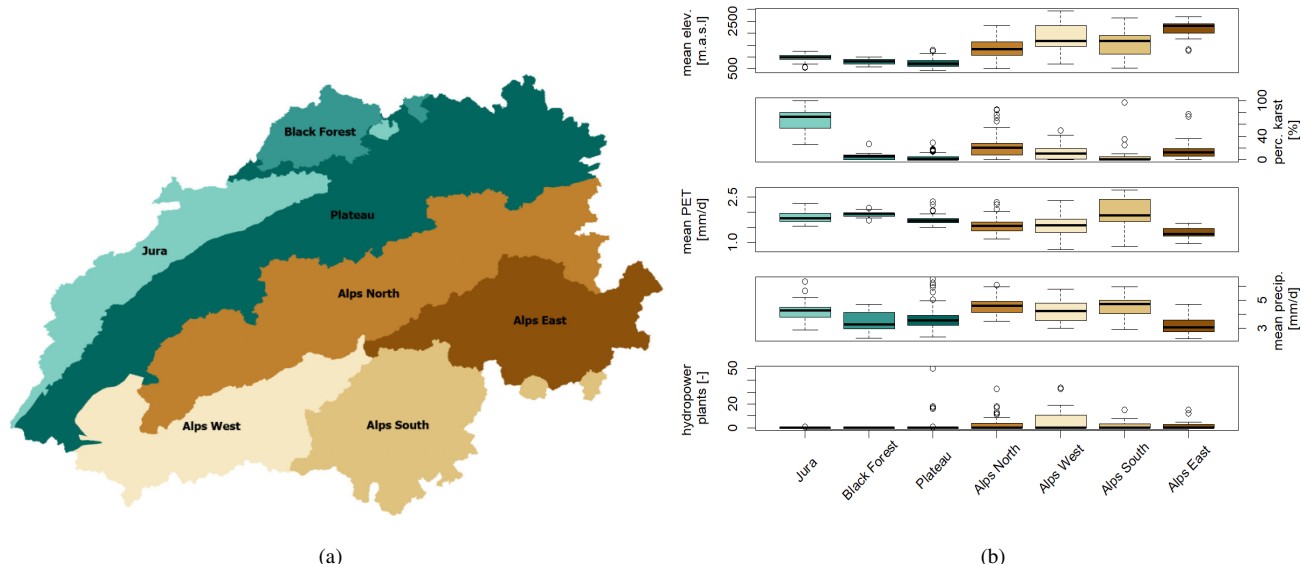

(a)                                                                          (b)

**Figure 4.** Bio-geographic regions of hydrologic Switzerland (a) with respective regional distributions of, from top to bottom, mean elevation,
percentage of karstified subsurface, mean potential evapotranspiration (PET), mean precipitation and the number of hydropower plants
$> 300$kW capacity in (b). Layout inspired by Figure 2 in Coxon et al. (2020)

.

Figure 4 (b) depicts that, unsurprisingly, the regions in the North-West (Jura, Black Forest, Plateau) have a much lower mean
elevation than the Alpine regions towards the South-East. Yet, catchments in valleys in the Northern, Western, and Southern
Alps reach similarly low elevation levels highlighting that there are large altitude differences in some of the Alpine basins. Note
that the climate attribute fraction of precipitation falling as snow (frac_snow; not shown here) highly correlates with the mean
elevation, with considerably more than $50\%$ of the annual precipitation falling as snow in some Alpine catchments. The Jura
basin is famous for its carbonate rock formations showing very high percentages of karstified subsurface in some catchments
which has to be considered for any hydrologic analysis in this region. Mean potential evapotranspiration (PET) appears to be
rather similar in the low-altitude regions and in the northern Alps. Yet, within the western and southern Alps, catchments cover
the entire range of low to high mean PET rates with the overall highest PET levels occuring in the South (Swiss Ticino and
northern Italy). The mountainous eastern Alps basins generally have the lowest PET rates in hydrologic Switzerland.. Mean
precipitation shows that there is overall less rain and snow in the Eastern Alps compared to all other Alpine regions while the
lower elevation catchments in the North-West are more similar, with only catchments in the Jura region showing higher values.





Finally, most hydropower capacity is installed in the regions of the South-West to North-East diagonal (Plateau, Alps North,
Alps West).

### 7.4    Hydrometry and uncertainty in hydrologic time series

Uncertainty in observed discharge and water level time series is caused by, e.g., the technique used to record discharges or
water levels, the extrapolation beyond the recording range, non-stationary conditions such as seasonal changes in vegetation or
in sediment erosion. Usually, discharge is not directly measured but calculated via rating curves from water level time series
that are much easier to measure at a high temporal resolution (Sikorska and Renard, 2017; McMillan et al., 2022). Rating
curves must be established for each gauging station independently using pairs of water level and discharge records taken
simultaneously directly at a river gauge. They allow to convert continuous water level records into *pseudo*-observed discharge
but are themselves subject to uncertainty Kiang et al. (see, e.g., 2018), in particularly for the range of low and high flows
(Westerberg et al., 2016).
CAMELS-CH comprises both observed discharge and water level time series (see Table 1). Uncertainty estimates are pro-
vided for exemplary catchments in order to inform respective analyses, but were not evaluated for each specific catchment. In a
recent study on a limited number of Swiss catchments, Westerberg et al. (2022) have investigated these uncertainty sources in
three Swiss catchments (gauge_ids: 2034, 2450 and 2469) of different properties and regime types (one glaciated, i.e. gauge_id:
2469). Using historical rating curves used by FOEN and 150 water level-discharge pairs measured between 1980 and 2014,
their uncertainties were estimated based on Monte Carlo sampling with the Voting Point likelihood method (McMillan and
Westerberg, 2015). Uncertainty in discharge estimates was calculated for selected low, medium and high flow quantiles, yield-
ing half-widths of the 5-95% uncertainty bounds for hourly $Q10$, $Q80$ and $Q99.9$ of on average $\pm22$-30%, $\pm10$-12% and
$\pm8.6$-20%, respectively (see Table 2. Yet, due to a lack of data pairs in the high-flow range corresponding estimates are vague.
High-flow events occur rarely and gauging them accurately is extremely challenging. Hence, in a companion study, Staudinger
and Viviroli (2020) refined the estimates for high flow quantiles, showing that the half-widths of the 2.5-97.5% uncertainty
bounds for hourly $Q99$, $Q99.9$ and $Q99.99$ data were on average $\pm14$%, $\pm15$% and $\pm22$%, respectively. This analysis was
based on the same three gauging stations and eleven others (gauge_ids: 2034, 2102, 2104, 2159, 2160, 2176, 2179, 2369, 2378,
2425, 2434, 2449, 2450, 2469) using the same method as in Westerberg et al. (2022).
Such rating curve uncertainties might further propagate to simulations of hydrological models if these were calibrated with
respective data (Westerberg et al., 2022), which has to be taken into account when working with the PREVAH-generated
discharge simulations here. Of course, also the meteorologic time series are subject to uncertainty. These are further specified in
the referenced data sources documentations in Sections A1.2 and A1.3, for observation and simulation-based data, respectively.

## 8    Conclusions

CAMELS-CH is the first freely available data set that covers the entirety of hydrological Switzerland, providing data for 298
catchments and 33 lakes. The data set comprises hydro-meteorologic daily time series as well as annual time series of land

cover and glacier data for the time between 1980 and 2020. Further, it contains catchment delineations and properties, including data on glaciation, reservoirs and hydropower that all play important roles in the Alpine water cycle.

Opportunities arise for eliciting long-term trends over decades for climate research for hydrological Switzerland or for conducting short-term local water cycle analyses in particular basins. Investigating human impacts regarding land cover or
reservoir and hydropower utilization is made possible as well as research on various subsurface properties and their relation to natural water flow and human water demand. CAMELS-CH can serve as a benchmark data set for improved modeling or analysis tools within Switzerland and also for other mountainous and alpine regions around the globe.

Within Switzerland's climate adaption strategy, water plays a key role (see Lanz, 2021). Hence, aside from many application scenarios discussed above, CAMELS-CH is meant to contribute to further hydrological and climatic research in Switzerland
that supports decision processes on adapting the various natural and human-made parts of the water tower of Europe.

CAMELS-CH further bears the potential for extensions, e.g. (i) refining of the temporal resolution from daily to hourly, e.g. based on the hourly discharge database HydroCH by Kauzlaric et al. (2023) (https://doi.org/10.5281/zenodo.7691294, last access: 21. March 2023) who collected and homogenized data for hydrological Switzerland; (ii) including cantonal data from rivers and streams not monitored by the FOEN; (iii) adding water quality and chemical data as it became available recently
for the original CAMELS (US) data set (Sterle et al., 2022) and that would certainly contribute to water quality research in Switzerland; (iv) adding CAMELS-CH data to the hydrologic atlas of Switzerland (HADES: https://hydrologicalatlas.ch/, last accessed: 21. March 2023).

The similar format of all CAMELS data sets fosters ease of utilization and international standardization. As a direct step towards creating a global unified data resource, CAMELS-CH was integrated into the Caravan data set (Kratzert et al., 2023) as
contribution to the growing field of global hydro-meteorological research (see Höge et al., 2023, ; last access: 20. May 2023).

## 9 Code and data availability

CAMELS-CH and the Caravan extension are freely available at: https://10.5281/zenodo.7957061 (Höge et al., 2023). Following FAIR data principles, we reference original data files (see also Section A1) and provide a transparent protocol of the applied data processing whenever possible at: https://camels-ch.github.io/. This corresponds to the CAMELS-CH code repository at:
https://github.com/camels-ch/camels.

## Appendix A: Additional information on attributes

### A1 Additional online references

Additional information about data sources and corresponding URLs to used online resources are listed below.



### A1.1 Hydrologic discharge

Hydrologic discharge timeseries for catchments outside of political Switzerland available as download or received upon personal request (see Kauzlaric et al., 2023) from

- Austria: Office of the Federal State of Vorarlberg, Division of Water Management, Bregenz (VRB, 2020, incl. personal request) and 15 catchments from LamaH (Klingler et al., 2021), i.e. gauge_ids: 3001, 3004, 3006, 3007, 3008, 3009, 3012, 3014, 3015, 3019, 3023, 3028, 3031, 3032, 3033.

- France: French database of discharge measurements (BanqueHydro, 2020)

- Germany: State Agency for the Environment Baden-Württemberg – Hydrographic Service, Karlsruhe (LUBW, 2020) and Bavarian State Office for the Environment – Hydrographic Service, Munich (GKDB, 2020)

- Italy: Regional agency for environmental protection for the region Lombardia, Milano (ARPALombardia, 2020, incl. personal request) and Regional agency for environmental protection for the region Piemonte, Torino (ARPAPiemonte, 470 2020, incl. personal request)

### A1.2 MeteoSwiss data products

Data product documentations by MeteoSwiss (2023) for

- daily precipitation: https://www.meteoswiss.admin.ch/dam/jcr:4f51f0f1-0fe3-48b5-9de0-15666327e63c/ProdDoc_RhiresD. pdf (last access: 15. May 2023)

- daily absolute temperature: https://www.meteoswiss.admin.ch/dam/jcr:818a4d17-cb0c-4e8b-92c6-1a1bdf5348b7/ProdDoc_ TabsD.pdf (last access: 15. May 2023)

- daily relative sunshine duration: https://www.meteoswiss.admin.ch/dam/jcr:981891db-30d1-47cc-a2e1-50c270bdaf22/ ProdDoc_SrelD.pdf (last access: 15. May 2023)

### A1.3 PREVAH Manual

PREVAH (Viviroli et al., 2007) provides so-called intercepted evapotranspiration $EI$ (intercept_et_sim in Table 1 and intercept storage $\Delta SI$ (intercept_storage_sim in Table 1) to describe the water balance at the soil surface underneath a plant canopy. With precipitation $P$ (precipitation_sim in Table 1) entering the canopy, the available water amount for runoff generation $P_b$ at the soil surface (as defined in Equation 4.2-1 of the PREVAH model description below), writes as:

$$P_b = P - EI - \Delta SI \tag{A1}$$

Note that $\Delta SI$ is a state variable difference in mm that has to be calculated per day, while the other three variables are fluxes in mm d$^{-1}$. For illustration, see Figure 4.2-1 the PREVAH model description, i.e. in Part II of Viviroli et al. (2007).





### A1.4 Swiss lakes

Information about the regulation of Swiss lakes is provided via FOEN (2020b): Unregulated lakes in the data set are Baldeg-gersee, Bodensee (Lake Constance), Lauerzersee, Sarnersee and Walensee. Note that their status might change in the future,
e.g. a flood relief tunnel is scheduled for completion on Lake Sarnersee in 2024, while a regulating weir is not planned there for the time being.

### A1.5 Hydrogeology

The hydrogeologic map of Switzerland (1:500'000 geo.admin.ch, 2016) covers the largest part of hydrological Switzerland. The hydrogeological map of Germany ((1:250'000) GDK, 2019) was used as extension to the North and West (product descrip-
tion available at: https://www.bgr.bund.de/DE/Themen/Wasser/Projekte/laufend/Beratung/Huek200/huek200_projektbeschr.html; last access: 15. May 2023)

### A1.6 Bio-geographic regions of Switzerland

There are six bio-geographic regions within political Switzerland (FOEN, 2022). These regions were extended to neighbouring countries and the southern Black Forest was added as seventh region to cover the entire hydrologic Switzerland for CAMELS-
CH.

### A2 Comparison of hydrogeologic attributes between CAMELS-CH and CAMELS-GB

Hydrogeologic attributes in CAMELS-CH were derived from local hydrogeologic maps to attributes names keep close resemblance. Directly relating them to their corresponding attributes in CAMELS-GB can be done using Table A1.



**Table A1.** Hydrogeology attributes: Comparison between CAMELS-CH and CAMELS-GB (Coxon et al., 2020)

| CH Attribute name | CH Description | GB Description | GB Attribute name |
|---|---|---|---|
| unconsol_coarse_perc | well-permeable gravel in valley bottoms | significant intergranular flow- high productivity | inter_high_perc |
| unconsol_medium_perc | permeable gravel outside of valley bottoms, sandy gravel, medium- to coarse-grained gravel | significant intergranular flow- moderate productivity | inter_mod_perc |
| unconsol_fine_perc | loamy gravel, fine- to medium-grained debris, moraines | significant intergranular flow- low productivity | inter_low_perc |
| unconsol_imperm_perc | clay, silt, fine sands and loamy moraines | unconsolidated material with essentially no groundwater | no_gw_perc |
| karst_perc | carbonate rock: limestone, dolomite, rauhwacke; sulphate-containing rock: gypsum, anhydrite | flow through fractures - high productivity | frac_high_perc |
| hardrock_perc | fissured and porous, non-karstic hard rock: conglomerates, sandstone, limestone with marl layers; crystalline rock: granite, granodiorites, tonalite | flow through fractures - moderate productivity | frac_mod_perc |
| | | generally not a significant aquifer but some low productivity (intergranular flow) | nsig_low_perc |
| | | flow through fractures - low productivity | frac_low_perc |
| | | generally low productivity (intergranular flow) but some not a significant aquifer | low_nsig_perc |
| hardrock_imperm_perc | marl, shale, gneiss and cemented sandstone | rocks with essentially no groundwater | no_gw_perc |
| null_perc | polygons without defined hydrogeology | – | – |
| water_perc | glaciers, firn, surface waters | water | – |





## Appendix B: Supplementary catchment attributes

For the three categories "location and topography" (see Section 6.1), "soil" (see Section 6.3) and "geology" (see Section 6.5), we provide supplementary attributes to provide a more comprehensive coverage and to account for some specifics of Switzerland: Regarding location and topography, note that the coordinate information on northing and easting above is provided in the Swiss reference system LV95. This is the coordinate system introduced in 1995. Yet, many public resources are still available in the old reference system LV03 (established in 1903). Hence, we additionally provide the old northing and easting

values in the supplementary topography file. Further, with many basins being nested and in order to account for the hierarchy between major rivers and minor streams and their tributaries, we provide two hierarchy numbers for bounding all connected stream parts. For example, the well-investigated Thur river (e.g., Abbaspour et al., 2007; Lopez et al., 2015; Doulatyari et al., 2017; Rössler et al., 2019; Dal Molin et al., 2020) with its length of 127 km and its catchment area (gauge_id: 2044) of $1702 km^2$ nests further basins and is itself part of larger ones. It has hierarchy_lower = 159 as lower hierarchy bound and hierarchy_upper

= 178 as upper bound. All catchments that have their own lower bound hierarchy_lower between these two numbers are nested within the Thur basin. In this case, these are two catchments (gauge_id: 2386 with bounds 160 and 163, and gauge_id: 2181 with bounds 164 and 177) with the first one nesting one further catchment and the second one several others.

As hydrological Switzerland covers several five countries, we also report the fractions of how much of each catchment area lies in the respective countries. We provide the areal percentage of each basin covered by the gridded meteorologic

data products that were used to aggregate the basin-specific time series for precipitation, temperature and relative sunshine duration (MeteoSwiss) as well as all simulated variables (WINMET). The catchments within political Switzerland are well-covered by all products but several basins in the neighbouring countries lack coverage by certain MeteoSwiss data. Note that for precipitation the spatial coverage was extended from political to hydrological Switzerland in 1992. Therefore, some catchments have a ext_r1981 of zero but are well covered for ext_r1992. Yet, while observed precipitation is available for many basins,

some lack observation-based temperature and relative sunshine duration data. Yet, simulated data time series can be used to fill such gaps.

Supplementary attributes for soil are derived from the global SoilGrid database (Hengl et al., 2017) and more detailed geological attributes were obtained Swiss data sources, i.e. from the Swiss Geological Map (map.geo.admin.ch, 2021) and the GeoMol database (GeoMol, 2021). The respective variables (geo_OSM, geo_OMM,...) provide greater detail with respect

to geologic formations across the country. While the Swiss mountainous regions are dominated by crystalline, sedimentary (central Alps) and karstified formations (north of the Swiss Plateau) the Swiss Plateau is dominated by different layers of Molasse. These Molasse layers differ in origin (marine and freshwater sources) and in their physical properties. These were shown to be good predictors of hydrological behavior in previous studies (e.g., Carlier et al., 2018; Floriancic et al., 2022).

**Table B1.** Supplementary catchment-specific static attributes (*Suppl. soil attributes: each soil type/property is accompanied by percentiles (5, 25, 50, 75, 90), distribution skewness and missing percentage across the catchment)





| Attribute class | Attribute name | Description | Unit | Data Source |
|---|---|---|---|---|
| Suppl. location and topography | gauge_id | catchment identifier according to FOEN notation, adjusted to neighbouring countries | – | Swiss Federal Office for the Environment (FOEN, 2023) / Swiss Federal Office of Topography (swisstopo, 2022) |
| | gauge_easting_old | gauging station easting in the old Swiss coordinate system LV03 | $m$ | |
| | gauge_northing_old | gauging station northing in the old Swiss coordinate system LV03 | $m$ | |
| | hierarchy_lower | catchment hierarchy lower bound | – | |
| | hierarchy_upper | catchment hierarchy upper bound | – | |
| | river_basin | large pan-European river basin that drains the catchment | – | |
| | river_basin_ch | pan-Switzerland river basin that drains the catchment | – | |
| | sub_gauge_id | | – | |
| | sub_area | | $km^2$ | |
| | frac_ch | catchment fraction within Switzerland | – | |
| | frac_de | catchment fraction within Germany | – | |
| | frac_at | catchment fraction within Austria | – | |
| | frac_li | catchment fraction within Liechtenstein | – | |
| | frac_it | catchment fraction within Italy | – | |
| | frac_fr | catchment fraction within France | – | |
| | ext_r1981 | catchment fraction covered by gridded rainfall data from 1981 until 1991 | % | Swiss Federal Office of Meteorology and Climatology (MeteoSwiss, 2023) |



Table B1 – continued from previous page

| Attribute class | Attribute name | Description | Unit | Data Source |
|---|---|---|---|---|
| | ext_r1992 | catchment fraction covered by gridded rainfall data from 1992 until 2020 | % | |
| | ext_tabs | catchment fraction covered by gridded temperature data | % | |
| | ext_srel | catchment fraction covered by gridded relative sunshine duration data | % | |
| | ext_sim | catchment fraction covered by the simulation-based meteorologic variables | % | PREVAH model (WINMET tool; Viviroli et al., 2007) |
| Suppl. soil* | sg_sand_perc | percentage sand | % | SoilGrids (2020) |
| | sg_silt_perc | percentage silt | % | |
| | sg_clay_perc | percentage clay | % | |
| | sg_organic_perc | percentage organic material | % | |
| | sg_bulk_dens | bulk density | g cm$^{-3}$ | |
| | sg_tot_avail_water | total available water content | mm | |
| | sg_root_depth | depth available for roots | m | |
| | sg_soil_depth | depth of soil | m | |
| Suppl. geology | geo_osm | Upper Freshwater Molasse | % | Swiss Geological Map (map.geo.admin.ch, 2021) / Geological 3D model of the Swiss Plateau (GeoMol, 2021) |
| | geo_omm | Upper Marine Molasse | % | |
| | geo_usm | Lower Freshwater Molasse | % | |
| | geo_api | Autochtone - Parautochtone, Infrahelvetic Nappes | % | |





Table B1 – continued from previous page

| Attribute class | Attribute name | Description | Unit | Data Source |
|---|---|---|---|---|
| | geo_amk | Allochthone Massive and Infrapenninic Crystallin Nappes | % | |
| | geo_sus | South- to Ultrahelvetic Sedimentary Nappes & Scales | % | |
| | geo_dup | Nappes of the Lower Eastalpine-Penninic Boundary Zone | % | |
| | geo_sal | Southalpine | % | |
| | geo_oos | Ophiolite containing Upperpenninic Sedimentary Nappes & Scales | % | |
| | geo_mpk | Midpenninic Crystallin Nappes | % | |
| | geo_aap | Outeralpine Plateau | % | |
| | geo_fju | Folded Jurassic | % | |
| | geo_hes | Helvetic Sedimentary Nappes s.str. | % | |
| | geo_mps | Midpenninic Sedimentary Nappes & Scale | % | |
| | geo_ups | Lower Penninic Sedimentary Nappes & Scale, Ophiolites | % | |
| | geo_ops | Upperpenninic Sedimentary Nappes | % | |
| | geo_tie | Tertiary Intrusives und Extrusives | % | |
| | geo_ukd | Lower Penninic Crystallin Nappes | % | |
| | geo_uod | Lower Eastapline Nappes | % | |
| | geo_ood | Upper Eastapline Nappes | % | |
| | geo_qua | Quarternary Deposits | % | |



*Author contributions.* M.H. initiated and managed the CAMELS-CH project and corresponding online repositories, prepared and processed
data, created most figures and mainly wrote the manuscript. M.K., R.S., U.S., P.H. and J.S. were part of the core team that prepared and
processed data, created maps, maintained the Github repository and documentation, and supported the manuscript writing. All other authors
suggested improvements and made additions to the manuscript, provided data and expertise for specific topics: M.G.F. (geology), D.V.
(hydrogeology), S.W. (glaciers outside of Switzerland), A.E.S.-S. (hydrometry), M.B. (reservoirs and human impact), S.P. (hydrology and
climate), M.Z. (PREVAH simulations). F.F. and M.H. acquired funding. N.A. and F.F. advised the project. All authors contributed to finalizing
the manuscript.

*Competing interests.* The authors do not declare any conflict of interest.

*Acknowledgements.* We thank the different Swiss institutions FOEN/BAFU, MeteoSwiss, swisstopo, SLF (Tobias Jonas) and GLAMOS
(Matthias Huss) for providing observation-based data and expertise. Further, we acknowledge the Copernicus programme https://www.
copernicus.eu/en as great resource and thank the data-providing agencies in Austria, France, Germany and Italy for their support. We want to
express our gratitude to Bettina Schaefli (UBern, head of the Swiss hydrologic commission), Peter Reichert (emeritus, former Eawag), Jan
Seibert (UZürich), Paolo Burlando (ETH Zürich), Paolo Benettin (EPFL), James Kirchner (ETH Zürich) and Christian Stamm (Eawag) for
supporting the CAMELS-CH initiative from an early stage on.



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
