# Peer review of "CAMELS-CH: hydro-meteorological time series and landscape attributes for 331 catchments in hydrologic Switzerland"

_Earth System Science Data, 2023_

## Author Response (AR1)

We thank two reviewers for their positive evaluation and feedback. Please find our replies below:

**Comment 1 (anonymous):**

The study from Marvin et al. presents CAMELS dataset for 331 basins in hydrological Switzerland. It's pleased to see more and more publicly available data from different countries, which has laid an important foundation for large-scale hydrological analysis. It's also great to see the efforts from authors to collect data for catchments in hydrological Switzerland including the one from neighbouring countries. Also, it is quite novel to include glacier and land cover annual data into the CAMELS datasets. Overall, I think the paper is suitable for Earth System Science Data journal. There are several places that could be improved before publication, I would recommend minor revision. My detailed comments can be found below.

*Thank you very much for your positive feedback and helpful remarks below.*

(1)Figure 1a: the river name label in light blue is too small to see.

*We increased font size and specified the legend of the map for clarity.*

(2)Line140-145: it would be better to add the median size of all catchments.

*Since the dataset contains a number of nested catchments, we initially decided to only provide the range of catchment sizes together with Figure 1 (b) to provide visual information on the matter. Yet, we agree that reporting the median in addition gives readers a better idea of the shape of the distribution of areas in the dataset. Therefore, we added it as suggested.*

(3)Line 180: as for the sentence: 'The observed time series data above are complemented by simulation-based time series products to bridge the spatial discrepancy between political and hydrological Switzerland, to fill temporal gaps in observation-based time series…', can you make it clear which variables of observed times series data are used simulated data as supplement? Since simulation-based variables might contain uncertainties, I would suggest to include one indicator (like quality control code) for observed time series data to indicate which data for specific time periods are complemented from simulated data. Because from your descriptions, the observations might use simulated data as supplement to fill the temporal gaps. The key point is to make it very clear whenever the data are complemented from simulated time series. Or I think it would be better to also have an observation only time series data and not fill the temporal gaps from simulated data. It's ok to have missing values in observation data and you can provide simulation-based time series data as reference. The users might need to make their own decision whether they want to use simulated data to fill the gap or not.

*Thank you for highlighting this as potential source of confusion. We did not combine any observed with simulated data to fill gaps, etc. We only discuss this as a potential option for users. Now, we created a new paragraph (l.178ff) that provides more information, clarifying this by stating "Note that observation-based and simulation-based data series are clearly separated in the data set, e.g., no gaps in observed time series were closed using simulated data. Any joint utilization is up to the user and we recommend a clear indication of how the different data are used."*

(4)It's nice to see authors have included annual glaciers and land cover into dataset. Especially the annual changes of land cover are useful for analysing human-nature

interactions for catchments. In terms of land cover product, is there any particular reason for using CORINE Land Cover (CLC) data sets? Have you try some other land cover datasets? Because I find the temporal resolution of CLC datasets might not ideal, as the data have quite long temporal interval and lots of interpolation work need to be done. The below link includes a global land cover datasets at 30 m resolution for the period 1982-2021, which might be good an alternative to try.

https://www.tandfonline.com/doi/full/10.1080/15481603.2022.2096184

*Thank you for highlighting this alternative resource. We are aware that a higher temporal resolution might sometimes be beneficial. Yet, in the area we covered, there have not been drastic changes so we think the applied linear interpolation is appropriate. For the European area, the CLC products are very reliable and of high quality. The referenced resource further provides significantly less categories of land cover features – presumably because of its global focus. Therefore, we settled with CLC and clearly documented which aggregation and interpolation steps were applied.*

(5)Line 305-310: Is there a particular reason you compare the hydrogeology data with CAMELS-GB? Why you choose CAMELS-GB as a reference? Why not compare with other CAMELS dataset, i.e. CAMELS-US etc.?

*CAMELS-GB was the second CAMELS dataset after the original US version and it partially provided new standards according to what was learned from the US dataset. This, and the fact that the available data resources in Switzerland covered many attributes that were also covered in GB, motivated us to take CAMELS-GB as comparable reference. Further, CAMELS-GB specifically provides "Hydrogeological attributes" as category while CAMELS-US more generally provides porosity and permeability in a Table of "Geological characteristics". In order to enable a direct comparison between CAMELS-GB hydrogeologic attributes and those that we provide in CAMELS-CH, we decided to provide Section A2 Comparison of hydrogeologic attributes between CAMELS-CH and CAMELS-GB.*

(6)I notice that in the csv format data, all variables for one timestep (time series data) or for one catchment (static attributes) are in one Excel-Cell and separated by a semicolon at the moment. Would that be possible to provide data that already splits up into different columns, which will be more friendly for users?

*Yes, this is possible. The second reviewer raised the same point. We decided to use semicolon because in German notation, commas are used to write decimal numbers. Since we expected a lot of German-speaking users, we wanted to avoid confusion here. Yet, we are fine with adopting to the international standard of using commas as separator and changed it.*

(7)The link in session '9 Code and data availability' for CAMELS-CH is not working. It might be due to missing 'doi.org' in the link.

*Thank you for highlighting this unfortunate bug. There might have been a copy-paste error. We fixed it.*

**Comment 2 (Rosanna Lane):**

**Overview**

This paper presents CAMELS-CH: a dataset of hydro-meteorological time-series and catchment attributes for catchments across hydrological Switzerland. This complements the existing CAMELS datasets, and will facilitate large sample studies.

I first want to say that I think it is fantastic that the authors have compiled this dataset, and I appreciate the large amount of effort that has gone into gathering and formatting many different data sources. The previous CAMELS datasets have proved very useful to the community, and it is excellent that Switzerland is being added. The list of time-series variables and catchment attributes presented is comprehensive, and the annual time-series of glacier attributes and land cover are valuable additions. The paper itself is well-written and logically structured, with high quality figures throughout. I particularly like that the authors have discussed and plotted the novel features of the dataset (changing glacier coverage and land use over time), and the plot showing the bio-geographic regions of Switzerland (Fig. 4) is very useful for those not familiar with Swiss hydrology. It is also helpful that the authors have used the same climatic and hydrologic attributes and corresponding descriptions as in CAMELS-GB, as this will assist future large sample studies that are using both datasets. I therefore would recommend that this manuscript is accepted into ESSD, but suggest the following minor revisions.

*Thank you very much for your positive feedback and helpful remarks below.*

**Minor comments:**

The zenodo link to the dataset provided in section 9 did not work, but the link within the abstract worked fine. Please make sure to double check all links within the manuscript.

*Thank you for highlighting this unfortunate bug. There might have been a copy-paste error. We fixed it.*

(2) Section 4: It looks from Figure 1 that the 331 selected gauging stations have good coverage across the country, but I was curious how/why you selected these stations. In particular, 1) were the 298 river catchments and 33 lakes all of the available data?, 2) did you carry out any quality checks?, 3) were there any criteria (e.g. length of record, suitability of gauging station) that you used to filter out unsuitable gauging stations?

*As mentioned by the reviewer, our goal was to have a good spatial coverage of hydrologic Switzerland.*

1) *For the 102 rivers/streams in neighboring countries (Austria: 33, France: 32, Germany: 26, Italy: 11) we included what we could obtain even if the temporal coverage within 1981 and 2020 is incomplete. Within political Switzerland (196 rivers/streams and 33 lakes), we collaborated with FOEN and therefore only provide federal data. Cantonal data was excluded as this stage because of varying standards between different cantons and to avoid potential inconsistencies.*
2) *We inspected the available data series visually and conducted plausibility checks, e.g. checking for offsets in the data and realistic magnitues. Mainly we trust the data providing agencies that we received data from and that apply quality checks*

*themselves. For example, all discharge data released by the Swiss FOEN is validated. However, as we point out in Section 5.1, the validation was not yet complete for data between 1. January 2019 and 31. December 2020, i.e. the last two years of our dataset.*

3) *The main filter was availability of data. Gauging stations were excluded during ordering the data from agencies if they were known to be unsuitable beforehand. This was based on mutual exchange. Hence, we include even basins for which only a small fraction of the intended 40 years – this occurs for a few catchments from neighboring countries. Yet, our major priority was to cover political Switzerland and to add basins in neighboring countries for completeness. With own CAMELS initiatives in these countries or similar on the way, we hope for better availability from these resources.*

(3) Line 154: what was the motivation behind the selection of this time period?

*Radiation data was only available since 1981 and this data is needed for the ET estimation. In order to provide a large list of time series variables that is comparable to other CAMELS datasets we started from this year. The end year 2020 was set since more recent data have not been fully validated by some of our data providers to be publicly released. Therefore, we decided to complete 40 years of data and end in 2020.*

(4) Line 192: not just simulation-based variables: observation based variables are also subject to uncertainties due to the interpolation method.

*We agree. For the discharge data, this is addressed in Section "7.4 Hydrometry and uncertainty in hydrologic time series", for the snow-water equivalent data, we have discussed and referenced respective information in the corresponding paragraph of Section 5.1 (l. 170ff). Now, we added a sentence to the paragraph before that refers to the variables provided by MeteoSwiss. There, we reference the additional material for meteorological data linked in the Appendix (A1.2) that also includes information about uncertainties of the data and used interpolation methods.*

(5) Line 286: I agree that it makes sense to only include data for hydrological years with 95% or more records. Is there a variable somewhere that says how many years were included in the calculation of observed hydrological signatures? It would be useful to at least highlight catchments where calculations are based on relatively few years and therefore may have less robust hydrological signatures. [I've now seen that this is mentioned in the next paragraph – it may be worth moving the text 'the start date, end date and number of years used for the calculations' earlier to L286/7].

*We agree and we changed the order of the paragraphs.*

(6) Line 291: It would be helpful to add the years covered by CAMELS-GB here as well

*We added the years "(1970-2015)" covered by CAMELS-GB.*

(7) Figure 1: I found it hard to pick out the yellow dots against the grey (I had to zoom in quite a lot to this figure). A border around the dots might help make them more easily visible.

*We agree. Thank you for the suggestion. We tried a few options and found that especially for the smallest dots adding a margin limited visibility of the color code. Therefore, we shifted the color scale to use colors with more contrast w.r.t. the background. We hope that the reviewer can follow our reasoning here and that visibility has improved.*

(8) Figure 3: It is difficult to directly compare the observed and simulated data in map form. It would be helpful to additionally show the data in a way that allows direct comparisons: for example, scatter plots (observed vs simulated), distribution plots such as CDFs (with separate lines for observed and simulated) or maps of the differences.

*We agree. As suggested, we added scatter plots and cdf plots in an appendix chapter for further inspection. We know that there are differences between observation-based and simulation-based signatures/indices and discuss in Section 7.2 that simulated values might be used as substitutions/additions – but that this is up to the users to which extend and whether it is suitable for their problem. Hence, we keep the analysis short and only show some signatures/indices as examples. We think that this analysis is important and has now a good extend that still lies within the scope of a data description paper. Anything beyond would be an analysis between observations and simulations.*

(9) The data were generally intuitively labelled and formatted. However, I noticed that that some files contained both NA and NaN values (e.g. CAMELS_CH_obs_based_6010.csv). It would be easier for users if these were kept consistently as NaN values.

*Thank you. We fully agree and noticed that this was the case for catchments outside of political Switzerland. We curated all respective files such that now it is only NaN everywhere.*

(10) I noticed that the .csv files have data separated by a semi-colon, meaning that they all appear in a single column when opened in Excel. It might be better to separate data using a comma so that it is more intuitive and easy to quickly look at the data. However, this is not a major issue.

*We agree. The first reviewer raised the same point. We decided to use semicolon because in German notation, commas are used to write decimal numbers. Since we expected a lot of German-speaking users, we wanted to avoid confusion here. Yet, we are fine with adopting to the international standard of using commas as separator and changed it.*

**Typos:**

Line 191: missing .

Line 396: extra .

Line 416: missing )

*Thank you for spotting these typos, we corrected all of them.*

**Further changes:**

- *We corrected small typos in the text, and added spaces between numbers and units where missing.*
- *In CAMELS_CH_topographic_attributes.csv, we found a typo that we corrected (also in the catchment delineation shape files):*

  *2247;CH;Sortie_du_lac_des_Brenets;Douds → now it is «Doubs »*

- *There are two gauged streams that connect close-by lakes. Depending on the water level in the lakes, flow might change direction in these streams. Therefore, the discharge time series have negative numbers. We added a small explanation about this in the readme file, highlighting that these two gauge_ids shall be excluded for rainfall runoff analysis:*

  *"gauge_ids 2446 (Zihlkanal) and 2447 (Canal_de_la_Broye), both at Lac de Neuchâtel: discharge timeseries partially comprise negative values as both refer to channels with bidirectional flow between lakes. Depending on the water level changes in these lakes, the flow direction in the channels might change as indicated by a positive or negative values in the time series."*

- *We added an extended explanation and table entries about "sub_gauge_id" and "sub_area" in Appendix B: Supplementary catchment attributes. Correspondingly, this update in the table entries was applied to in the data description file in the Zenodo repository.*
- *There was an error in the previous aggregation of the simulation-based precipitation and evapotranspiration data. The corresponding time series as well as related hydrology and climate attributes were corrected in the Zenodo repository. The analysis of observation-based vs. simulation-based runoff ratios in Section 7.2 changed and was rewritten accordingly.*
- *All changes in the Zenodo repository are documented in the version updates on the repository webpage.*